# Household transmission of SARS-CoV-2 in five US jurisdictions: Comparison of Delta and Omicron variants

**Julia M. Baker**[1,2]*, **Jasmine Y. Nakayama**[1,2], **Michelle O'Hegarty**[1], **Andrea McGowan**[1,3], **Richard A. Teran**[2,4], **Stephen M. Bart**[2,5], **Lynn E. Sosa**[5], **Jessica Brockmeyer**[5], **Kayla English**[4], **Katie Mosack**[6¤], **Sanjib Bhattacharyya**[6], **Manjeet Khubbar**[6], **Nicole R. Yerkes**[7], **Brooke Campos**[7], **Alina Paegle**[7], **John McGee**[7], **Robert Herrera**[7], **Marcia Pearlowitz**[8], **Thelonious W. Williams**[8], **Hannah L. Kirking**[1], **Jacqueline E. Tate**[1]

**1** COVID-19 Response Team, Centers for Disease Control and Prevention, Atlanta, Georgia, United States of America, **2** Epidemic Intelligence Service, Centers for Disease Control and Prevention, Atlanta, Georgia, United States of America, **3** Oak Ridge Institute for Science and Education, Oak Ridge, Tennessee, United States of America, **4** Chicago Department of Public Health, Chicago, Illinois, United States of America, **5** Connecticut Department of Public Health, Hartford, Connecticut, United States of America, **6** Milwaukee Health Department, Milwaukee, Wisconsin, United States of America, **7** Utah Department of Health and Human Services, Salt Lake City, Utah, United States of America, **8** Maryland Department of Health, Baltimore, Maryland, United States of America

¤ Current address: Oregon Clinical and Translational Research Institute, Portland, Oregon, United States of America

* nwk0@cdc.gov

**Data Availability Statement:** The analyzed data include individual-level demographic, health, SARS-CoV-2 testing, and household information collected for public health surveillance purposes by

## Abstract

Households are a significant source of SARS-CoV-2 transmission, even during periods of low community-level spread. Comparing household transmission rates by SARS-CoV-2 variant may provide relevant information about current risks and prevention strategies. This investigation aimed to estimate differences in household transmission risk comparing the SARS-CoV-2 Delta and Omicron variants using data from contact tracing and interviews conducted from November 2021 through February 2022 in five U.S. public health jurisdictions (City of Chicago, Illinois; State of Connecticut; City of Milwaukee, Wisconsin; State of Maryland; and State of Utah). Generalized estimating equations were used to estimate attack rates and relative risks for index case and household contact characteristics. Data from 848 households, including 2,622 individuals (median household size = 3), were analyzed. Overall transmission risk was similar in households with Omicron (attack rate = 47.0%) compared to Delta variant (attack rate = 48.0%) circulation. In the multivariable model, a pattern of increased transmission risk was observed with increased time since a household contact's last COVID-19 vaccine dose in Delta households, although confidence intervals overlapped (0–3 months relative risk = 0.8, confidence interval: 0.5–1.2; 4–7 months relative risk = 1.3, 0.9–1.8; ≥8 months relative risk = 1.2, 0.7–1.8); no pattern was observed in Omicron households. Risk for household contacts of symptomatic index cases was twice that of household contacts of asymptomatic index cases (relative risk = 2.0, 95% confidence interval: 1.4–2.9), emphasizing the importance of symptom status, regardless of variant. Uniquely, this study adjusted risk estimates for several index case and household contact characteristics and demonstrates that few characteristics strongly dictate risk, likely

jurisdictional health departments or CDC. Data cannot be shared publicly because they contain certain identifiable information. Data may be available upon reasonable request from the CDC Coronavirus and Other Respiratory Viruses Division Informatics Team (corvdinformatics@cdc.gov) for researchers who meet the criteria for access to confidential data and with approval from participating jurisdictional partners.

**Funding:** This work was supported by the CDC's Epidemiology and Laboratory Capacity for Prevention and Control of Emerging Infectious Diseases: https://www.cdc.gov/elc/index.html [support provided to Maryland Department of Health; Utah Department of Health and Human Services]. CDC had a role in the investigation design, data collection, analysis, decision to publish, and preparation of the manuscript.

**Competing interests:** I have read the journal's policy and the authors of this manuscript have the below competing interests. This does not alter our adherence to PLOS ONE policies on sharing data and materials. LES: Support for the present manuscript from Centers for Disease Control and Prevention; Support for attending meetings and/or travel from Centers for Disease Control and Prevention, Council of State and Territorial Epidemiologists, Association of State and Territorial Health Officers; Participation on a Data Safety Monitoring Board or Advisory Board for Centers for Disease Control and Prevention. NRY: Grants or contracts from CDC's Epidemiology and Laboratory Capacity for Prevention and Control of Emerging Infectious Diseases (ELC) Cares, ELC PPP, ELC EED (Constituted hourly wages received through Utah Department of Health and Human Services' payroll system). BC: Grants or contracts from CDC's Epidemiology and Laboratory Capacity for Prevention and Control of Emerging Infectious Diseases (ELC) PPPHEA, ELC ED (Constituted hourly wages received through Utah Department of Health and Human Services' payroll system). AP: Grants or contracts from CDC's Epidemiology and Laboratory Capacity for Prevention and Control of Emerging Infectious Diseases (ELC) Cares, ELC PPP, ELC EED (Constituted hourly wages received through Utah Department of Health and Human Services' payroll system). RH: Support for the present manuscript from CDC's Epidemiology and Laboratory Capacity for Prevention and Control of Emerging Infectious Diseases (ELC) Cares (Constituted hourly wages and received through Utah Department of Health and Human Services' payroll system); Grants or contracts from ELC PPP and Immunization Cooperative Agreements (Constituted hourly wages and received through

reflecting the complexity of the biological and social factors which combine to impact SARS-CoV-2 transmission.

## Introduction

The context of SARS-CoV-2 household transmission has evolved over the course of the COVID-19 pandemic. Many household transmission studies conducted shortly after the emergence of SARS-CoV-2 were epidemiologically relatively simple with low baseline rates of previous infection, no SARS-CoV-2 vaccination, and often limited social interaction outside the home [1–3]. Increased incidence of natural infection, circulation of several SARS-CoV-2 variants, availability of multiple vaccines, and reduced social distancing subsequently added layers of complexity to such studies [4–6]. Thus, estimates of SARS-CoV-2 household transmission have varied by time, geography, population and study method [7].

Despite their variability, household SARS-CoV-2 transmission estimates represent an important source of SARS-CoV-2 transmission—close, sustained exposure often experienced among household members and others with whom extended time is spent—even during times of low community-level transmission [8]. Estimating SARS-CoV-2 household transmission risk across a range of times and locations deepens our understanding of transmission risk in real-world settings and how it has changed over time, which, in turn, may have important implications for policy and public health guidelines as communities continue to navigate COVID-19. Comparing household transmission rates by SARS-CoV-2 variant may provide relevant, updated information about risks and inform prevention strategies such as updates to isolation guidance and future vaccine strategies. Examining how the epidemiology of SARS-CoV-2 does and does not change through different variant periods also helps us understand collective and cumulative changes that occur over the course of the pandemic, i.e., from the time of the emergence of a novel virus and throughout years of increasing population-level immunity.

Using data from contact tracing and interviews conducted from November 2021 through February 2022 in five U.S. public health jurisdictions, this investigation examined differences in household transmission comparing the SARS-CoV-2 Delta and Omicron variants. Unadjusted and adjusted models were used to estimate secondary attack rates (ARs) and quantify the extent to which variants influenced transmission risk.

## Materials and methods

Households for analysis were identified from five U.S. public health jurisdictions (City of Chicago, Illinois; State of Connecticut; City of Milwaukee, Wisconsin; State of Maryland; and State of Utah) that volunteered to participate and that had sequencing results available for a portion of SARS-CoV-2 positive tests in their jurisdiction. Household identification and data collection occurred 1) retrospectively, in which previously collected data from the jurisdictional health department's standard COVID-19 contact tracing interview were extracted or 2) prospectively, in which residents were contacted and interviewed using an investigation-specific questionnaire (see Baker et al. [9] for details). Two jurisdictions (Maryland and Utah) contributed retrospective data while four jurisdictions (Chicago, Connecticut, Milwaukee, and Utah) contributed prospective data. Eligible households were those with at least one person who had a sequence-confirmed SARS-CoV-2 Delta or Omicron infection from October 2021 through February 2022. For retrospectively identified households, data from contact tracing

Utah Department of Health and Human Services' payroll system). MP: Support for the present manuscript from CDC's Epidemiology and Laboratory Capacity for Prevention and Control of Emerging Infectious Diseases (ELC) (Cooperative agreement funding supported staff to conduct routine disease investigations and data analysis). TWW: Support for the present manuscript from DC Grant: CDC's Epidemiology and Laboratory Capacity for Prevention and Control of Emerging Infectious Diseases (ELC) (Cooperative agreement funding supported staff to conduct routine disease investigations and data analysis).

interviews were used to identify household contacts within the residence. For prospectively identified households, the individual with the first sequence-confirmed SARS-CoV-2 test was contacted and that individual and each household contact they reported was asked to participate in a voluntary interview (verbal informed consent was obtained and documented on data collection forms; parents provided consent for minors). All interviews for retrospectively and prospectively identified households were conducted from November 1, 2021 through February 28, 2022. Data for retrospectively identified households were compiled between February 23, 2022 and January 30, 2023.

In this investigation, the index case was defined as the first person in a residence to have a recent positive SARS-CoV-2 nucleic acid amplification or antigen test (confirmed index case) or to experience symptoms of a COVID-19 infection [10] (probable index case). Any individual who spent one or more overnights in the same residence as the index case during the index case's potentially infectious period was considered a household contact. The potentially infectious period was defined as 2 days before the index case's positive SARS-CoV-2 test specimen collection date or symptom onset date (whichever occurred first, hereafter referred to as the "index case date") up to 10 days after onset. Households were excluded if the index case resided alone during their infectious period, if more than one index case was identified (e.g., multiple individuals had simultaneous positive SARS-CoV-2 tests), if multiple variants were detected among individuals in the household, if household contacts with a positive SARS-CoV-2 test or symptoms had a known SARS-CoV-2 exposure other than the index case, or if the residence was a congregate living setting.

Case status of household contacts was defined similarly to that of index cases. A confirmed household contact case was defined as a positive SARS-CoV-2 nucleic acid amplification or antigen test result within 14 days after the index case date. A household contact was classified as a probable case if they experienced COVID-19 symptoms [10] during the same 14-day period, but without a positive viral SARS-CoV-2 test confirmation.

Data were primarily collected through index case and household contact interviews and included index case and household contact demographics, SARS-CoV-2 testing and results, COVID-19 symptoms, previous SARS-CoV-2 infection, and COVID-19 vaccination history. Interview data were supplemented with jurisdictional vaccination and surveillance records where available. Vaccination status was defined relative to the index case date and categorized as unvaccinated (never received a COVID-19 vaccine), partially vaccinated (completed part of a COVID-19 vaccine series or completed the series <14 days prior), fully vaccinated (completed the primary series ≥14 days prior or received a booster dose <7 days prior), or boosted (received an additional vaccine dose beyond the primary series ≥7 days prior). A composite variable, "number of prior immunologic experiences," was created which combined previous infection and vaccination doses to estimate the number of times a person's immune system had been exposed to SARS-CoV-2 or the vaccine. The possible range of values was zero to four and could be any combination of infection and vaccination (e.g., an individual categorized as having two exposures could have received two doses of a COVID-19 vaccine and had no previous infection or could have had one dose of a COVID-19 vaccine and a previous infection). Those with unknown previous infection status based on self-report and available surveillance records were assumed to have no previous infection.

Descriptive analyses were conducted with data stratified by SARS-CoV-2 variant. The serial interval was calculated as the number of days between the index case date and onset of symptoms or positive test result in a household contact. A generalized estimating equations (GEE) approach using a Poisson distribution and exchangeable correlation structure was employed to explore the bivariate relationship between case status of household contacts and several index and household member characteristics while accounting for clustering by household.

Using point estimates and robust standard errors from the GEE models, ARs and 95% confidence intervals (CIs) among household contacts were estimated using marginal means. The same characteristics were further compared by estimating relative risks (RRs) and 95% CIs.

Index and household member characteristics were evaluated in a multivariable model to produce adjusted RRs. A GEE model was fit using all characteristics that were not strongly correlated with others (pairwise correlation rho < .60). To assess potential changes in the relationship between index case and household member characteristics by variant, a variant-characteristic interaction term was specified for each characteristic in the initial model one at a time. An expanded model was constructed including all characteristics from the initial model along with the interaction terms found to be significant at the α = 0.05 level (i.e. an interaction term was considered significant if the 95% confidence interval for any level of the interaction term did not cross the null) in the prior modeling step. The final model was developed by retaining variant, index case characteristics and household member characteristics and including only the interaction terms that remained significant (at the α = 0.05 level).

The primary analysis was limited to households in which case status was determined for at least two-thirds of household contacts. A sub-analysis was conducted with the above-described multivariable model rerun on a dataset restricted to households with retrospectively collected data to assess if risk estimates differed by data collection method.

Data were collected and managed using REDCap (version 11.1.8; Vanderbilt University) and analyzed in R (version 4.2.1; R Foundation). Collected data included potentially identifiable information (e.g. index case contact information and dates of infection) and all data were stored securely and only available to investigators for data entry and analysis. This investigation was reviewed by CDC, determined to be public health surveillance (not research), and was conducted consistent with applicable federal law and CDC policy.

## Results

A total of 1,241 households meeting initial inclusion criteria were interviewed. Of these, 227 households were excluded after determination that the index case lived alone or in a congregate setting (n = 78), identification of multiple index cases in the household (n = 101), known shared exposures among the index case and household contacts (n = 24), known exposures outside of the household to SARS-CoV-2 among household contact cases (n = 13), insufficient household information (n = 6), or multiple or unknown SARS-CoV-2 variants identified in the household (n = 5). An additional 166 households were excluded because of low participation (i.e. limited data collection and unknown case status) among household contacts. Data from the remaining 848 households (848 index cases and 1,774 household contacts) were analyzed (Table 1).

Approximately three-quarters (77.6%) of households were identified retrospectively. The largest proportion of households were in Maryland (65.8%), followed by Utah (13.8%), Connecticut (11.9%), and Chicago and Milwaukee (both 4.2%) (Table 1, Fig 1A). There were slightly fewer Delta households compared to Omicron households (43.8% Delta, 56.2% Omicron) and index case dates ranged from the weeks of October 25, 2021, through February 14, 2022 (Fig 1B). The median household size was 3 (Interquartile range, IQR: 2–4). The median time from index case date to index case interview was 6 days (IQR: 4–14).

Almost two-thirds (63.0%) of index cases were 18–64 years of age and half were female (51.1%, Table 1). Most index cases identified as White (51.3%) and non-Hispanic/Latino (83.6%). Over half of index cases were either fully vaccinated (40.7%) or boosted (15.0%). Of those who had received at least one vaccine dose, 42.1% received their last dose within the previous 5 months. Approximately 29.8% of household contacts were fully vaccinated and 13.8%

**Table 1.** Demographic and clinical characteristics of index cases and household contacts.

| Characteristic | Index case[a] (N = 848) | Household contact[b] (N = 1774) | Total (N = 2622) |
|---|---|---|---|
| **Jurisdiction** | | | |
| Chicago, Illinois | 36 (4.2%) | 71 (4.0%) | 107 (4.1%) |
| Connecticut | 101 (11.9%) | 235 (13.2%) | 336 (12.8%) |
| Maryland | 558 (65.8%) | 1148 (64.7%) | 1706 (65.1%) |
| Milwaukee, Wisconsin | 36 (4.2%) | 101 (5.7%) | 137 (5.2%) |
| Utah | 117 (13.8%) | 219 (12.3%) | 336 (12.8%) |
| **Data collection method** | | | |
| Prospective | 190 (22.4%) | 452 (25.5%) | 642 (24.5%) |
| Retrospective | 658 (77.6%) | 1322 (74.5%) | 1980 (75.5%) |
| **SARS-CoV-2 variant in the household** | | | |
| Delta | 371 (43.8%) | 786 (44.3%) | 1157 (44.1%) |
| Omicron | 477 (56.2%) | 988 (55.7%) | 1465 (55.9%) |
| **Age group (years)** | | | |
| 0–4 | 42 (5.0%) | 149 (8.4%) | 191 (7.3%) |
| 5–11 | 83 (9.8%) | 246 (13.9%) | 329 (12.5%) |
| 12–17 | 61 (7.2%) | 166 (9.4%) | 227 (8.7%) |
| 18–64 | 534 (63.0%) | 1036 (58.4%) | 1570 (59.9%) |
| ≥65 | 115 (13.6%) | 156 (8.8%) | 271 (10.3%) |
| Unknown | 13 (1.5%) | 21 (1.2%) | 34 (1.3%) |
| **Sex** | | | |
| Female | 433 (51.1%) | 917 (51.7%) | 1350 (51.5%) |
| Male | 415 (48.9%) | 790 (44.5%) | 1205 (46.0%) |
| Other/unknown | 0 (0.0%) | 67 (3.8%) | 67 (2.6%) |
| **Race** | | | |
| White | 435 (51.3%) | 768 (43.3%) | 1203 (45.9%) |
| Black/African American | 171 (20.2%) | 310 (17.5%) | 481 (18.3%) |
| Asian | 40 (4.7%) | 88 (5.0%) | 128 (4.9%) |
| Other/multiple races | 37 (4.4%) | 64 (3.6%) | 101 (3.9%) |
| Unknown | 165 (19.5%) | 544 (30.7%) | 709 (27.0%) |
| **Ethnicity** | | | |
| Non-Hispanic/Latino | 709 (83.6%) | 1205 (67.9%) | 1914 (73.0%) |
| Hispanic/Latino | 106 (12.5%) | 231 (13.0%) | 337 (12.9%) |
| Unknown | 33 (3.9%) | 338 (19.1%) | 371 (14.1%) |
| **COVID-19 vaccination status[c]** | | | |
| Boosted | 127 (15.0%) | 244 (13.8%) | 371 (14.1%) |
| Fully vaccinated | 345 (40.7%) | 528 (29.8%) | 873 (33.3%) |
| Partially vaccinated | 23 (2.7%) | 54 (3.0%) | 77 (2.9%) |
| Unvaccinated | 331 (39.0%) | 657 (37.0%) | 988 (37.7%) |
| Unknown | 22 (2.6%) | 291 (16.4%) | 313 (11.9%) |
| **Time since last COVID-19 vaccine dose (months)** | | | |
| Unvaccinated/partially vaccinated | 354 (41.7%) | 711 (40.1%) | 1065 (40.6%) |
| 0–1 | 61 (7.2%) | 148 (8.3%) | 209 (8.0%) |
| 2–3 | 86 (10.1%) | 123 (6.9%) | 209 (8.0%) |
| 4–5 | 46 (5.4%) | 67 (3.8%) | 113 (4.3%) |
| 6–7 | 140 (16.5%) | 195 (11.0%) | 335 (12.8%) |
| 8–9 | 95 (11.2%) | 118 (6.7%) | 213 (8.1%) |

*(Continued)*

**Table 1.**  (Continued)

| Characteristic | Index case[a] (N = 848) | Household contact[b] (N = 1774) | Total (N = 2622) |
|---|---|---|---|
| 10+ | 30 (3.5%) | 31 (1.7%) | 61 (2.3%) |
| Unknown | 36 (4.2%) | 381 (21.5%) | 417 (15.9%) |
| **Previous infection** | | | |
| No | 683 (80.5%) | 552 (31.1%) | 1235 (47.1%) |
| Yes | 19 (2.2%) | 41 (2.3%) | 60 (2.3%) |
| Unknown | 146 (17.2%) | 1181 (66.6%) | 1327 (50.6%) |
| **Number of prior immunologic experiences[d]** | | | |
| 0 | 329 (38.9%) | 645 (37.2%) | 974 (37.8%) |
| 1 | 46 (5.4%) | 93 (5.4%) | 139 (5.4%) |
| 2 | 292 (34.6%) | 549 (31.6%) | 841 (32.6%) |
| 3 | 153 (18.1%) | 226 (13.0%) | 379 (14.7%) |
| 4 | 6 (0.7%) | 9 (0.5%) | 15 (0.6%) |
| Unknown | 22 (2.6%) | 252 (14.2%) | 274 (10.5%) |
| **Case status** | | | |
| Confirmed | 818 (96.5%) | 667 (37.6%) | 1485 (56.6%) |
| Probable | 30 (3.5%) | 118 (6.7%) | 148 (5.6%) |
| Not a case | 0 (0.0%) | 895 (50.5%) | 895 (34.1%) |
| Unknown | 0 (0.0%) | 94 (5.3%) | 94 (3.6%) |

[a] The index case was defined as the first person in a residence to have a recent positive SARS-CoV-2 nucleic acid amplification test or antigen test or to experience symptoms of a COVID-19 infection.

[b] A household contact was defined as any individual who spent one or more overnights in the same residence as the index case during the index case's potentially infectious period. The index case's potentially infectious period was defined as 2 days before the index case's positive SARS-CoV-2 test specimen collection date or symptom onset date (whichever occurred first) through the date of the interview or up to 10 days after onset.

[c] Vaccination status was defined as of the index case date and was categorized as unvaccinated (never received a COVID-19 vaccine), partially vaccinated (completed part of a COVID-19 vaccine series or completed the primary series <14 days prior), fully vaccinated (completed the primary series ≥14 days prior or received a booster dose <7 days prior), or boosted (received an additional vaccine dose beyond the primary series ≥7 days prior).

[d] Number of prior immunologic experiences combined previous infection and vaccination doses to estimate the number of times a person's immune system had been exposed to SARS-CoV-2 or the vaccine. The possible range of values was zero to four exposures and could be any combination of infection and vaccination. Those with unknown previous infection status were considered to have no previous infection for this composite variable.

Percentages may not total to 100.0% due to differences in rounding.

were boosted. A small proportion (2.2%-2.3%) of index cases and household contacts had a known previous infection (Table 1).

Nearly half (44.3%) of household contacts tested positive and/or developed COVID-19 symptoms within 14 days of the index case date. The secondary AR was similar by variant (Delta AR = 47.0%, 95% CI: 42.8–51.6; Omicron AR = 48.0%, 95% CI: 44.1–52.1). The median serial interval for both Delta and Omicron was 4 days (IQR: 2–6) (Fig 2).

ARs (Fig 3A) and unadjusted RRs (Fig 3B) were generally similar when comparing Delta and Omicron transmission by index case characteristics with overlapping CIs. By age group, the highest attack rates were observed when the index case was 0–4 years of age (Delta AR = 71.5%, 95% CI: 63.5–80.5; Omicron AR = 59.2%, 95% CI: 46.3–75.6). The RR varied by index case age group within Delta households; transmission risk was substantially lower for all index case age groups when compared to index cases aged 0–4 years. A similar pattern was observed in Omicron households, though with slightly attenuated RRs. For both Delta and Omicron, the risk of transmission was approximately twice as high when the index case was symptomatic compared to asymptomatic (Delta RR = 2.1, 95% CI: 1.2–3.9; Omicron RR = 2.0,

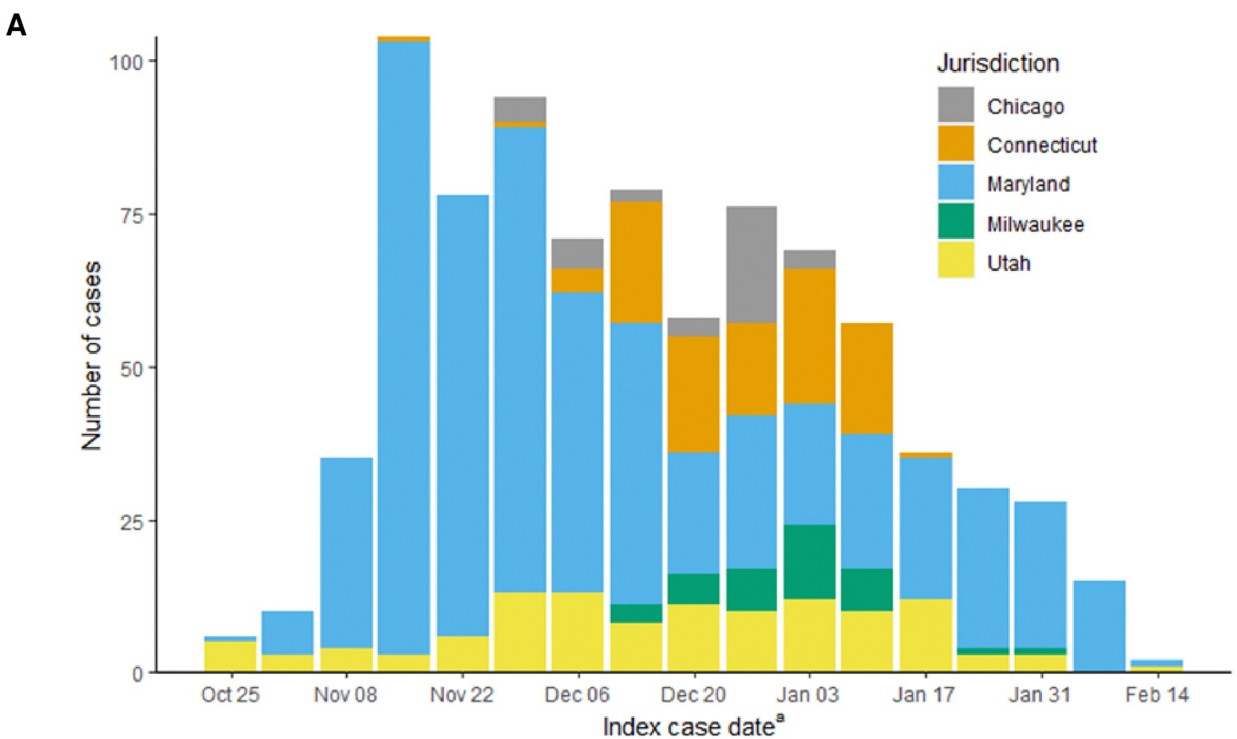

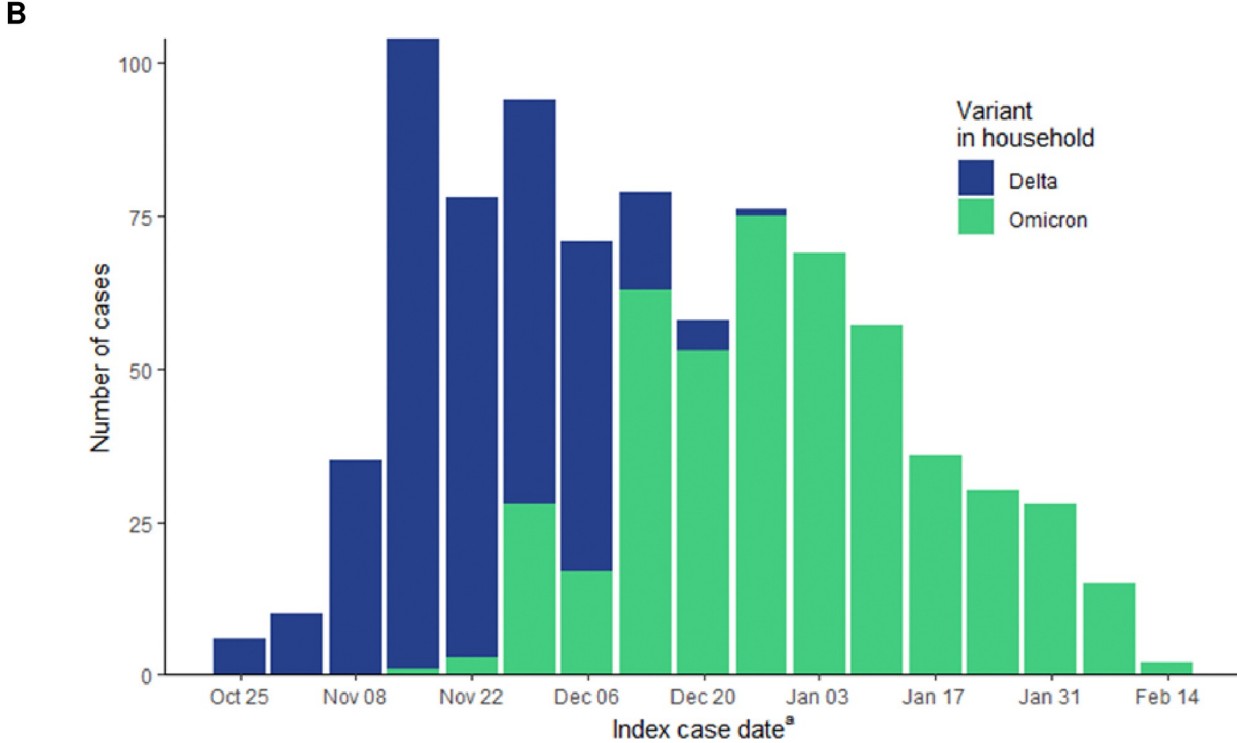

**Fig 1.** Index case date by A) jurisdiction and B) SARS-CoV-2 variant in the household for the weeks of October 25, 2021 –February 14, 2022. [a] Index case date is defined as the index case's positive SARS-CoV-2 test specimen collection date or symptom onset date (whichever occurred first).

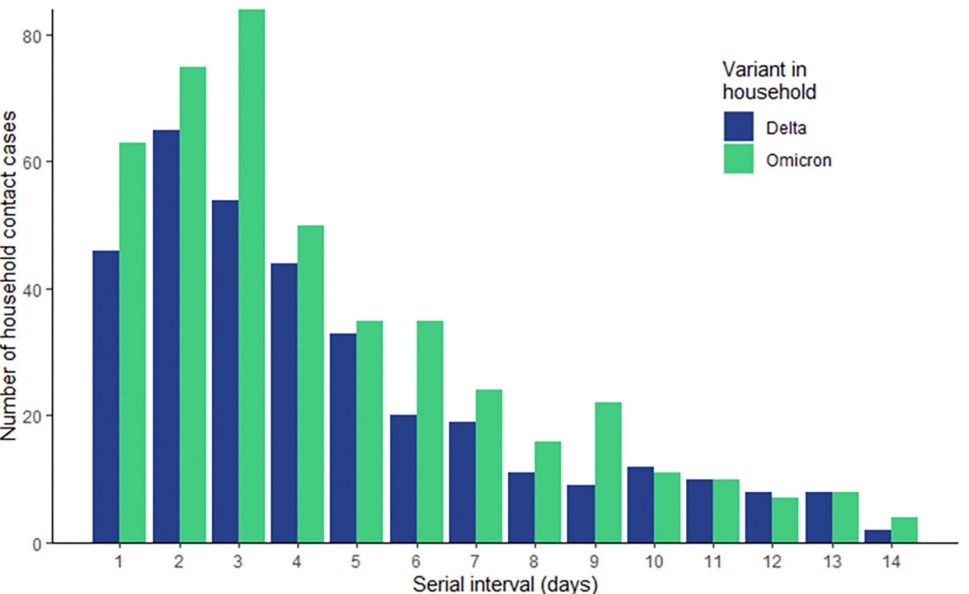

**Fig 2. Serial interval (days) between index case date[a] and household contact onset date[b].** [a] Index case date defined as the index case's positive SARS-CoV-2 test specimen collection date or symptom onset date (whichever occurred first). [b] Household contact onset date defined as the household contact's positive SARS-CoV-2 test specimen collection date or symptom onset date (whichever occurred first).

95% CI: 1.3–2.9). ARs were generally highest when the index case was unvaccinated (Delta AR = 49.3%, 95% CI: 44.0–55.4) or partially vaccinated (Omicron AR = 60.9%, 95% CI: 40.1–92.5). Excluding the 0-1-month period, a pattern of increasing ARs and RRs as the time since the index case's last vaccine dose increased was observed, particularly in Omicron households. Lastly, a pattern of increasing ARs and RRs was observed in Delta households as the number of prior immunologic experiences for the index case increased. The data suggested a subtler and opposite pattern in Omicron households.

The ARs (Fig 4A) and RRs (Fig 4B) in Delta households typically varied for different levels of household contact characteristics while those in Omicron households varied more modestly or demonstrated attenuated RRs. This pattern was most apparent for time since the household contact's last vaccine dose. Among household contacts in Delta households, a clear increase in ARs and RRs was observed as time since vaccination increased (AR range: 28.8%-76.1% and RR range: 0.6–1.6). Conversely, for household contacts in Omicron households, ARs were relatively similar across different time ranges (range: 45.0%- 59.0%) and the RR varied subtly (range: 0.7–1.2). In Omicron households, RRs for household contact age group, previous infection status, and vaccination status were all attenuated in comparison to Delta households and, with the exception of vaccination status, did not display strong patterns.

The initial multivariable model (S1 Table) included household variant, index case characteristics (age group, symptom status, vaccination status), and household contact characteristics (age group, time since last vaccine dose, number of prior immunologic experiences). Index case time since vaccination, index case number of prior immunologic experiences, and household contact vaccination status were excluded due to high correlation with other characteristics. The only significant interaction term for inclusion in the final model was household contact time since last vaccine dose (S1 Table, Model 5).

In the adjusted final model (Table 2), household variant was not associated with transmission risk (RR = 1.1, 95% CI: 0.9–1.3). The strongest predictor of transmission was index case

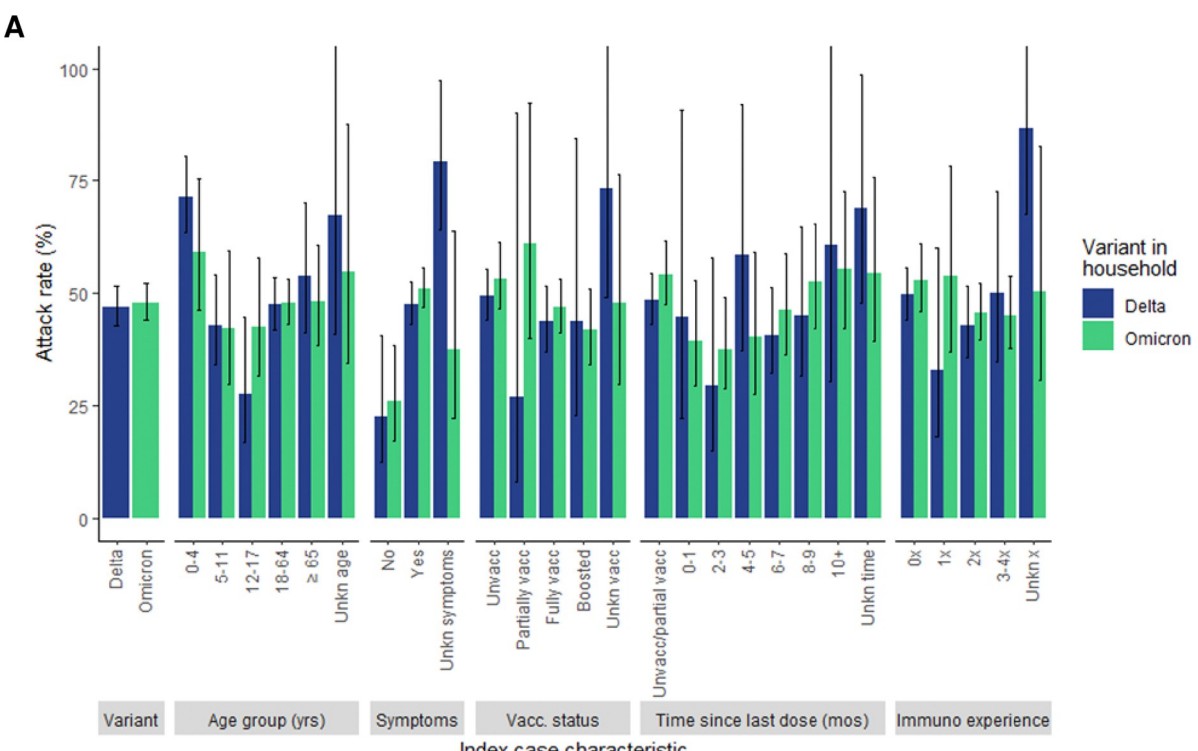

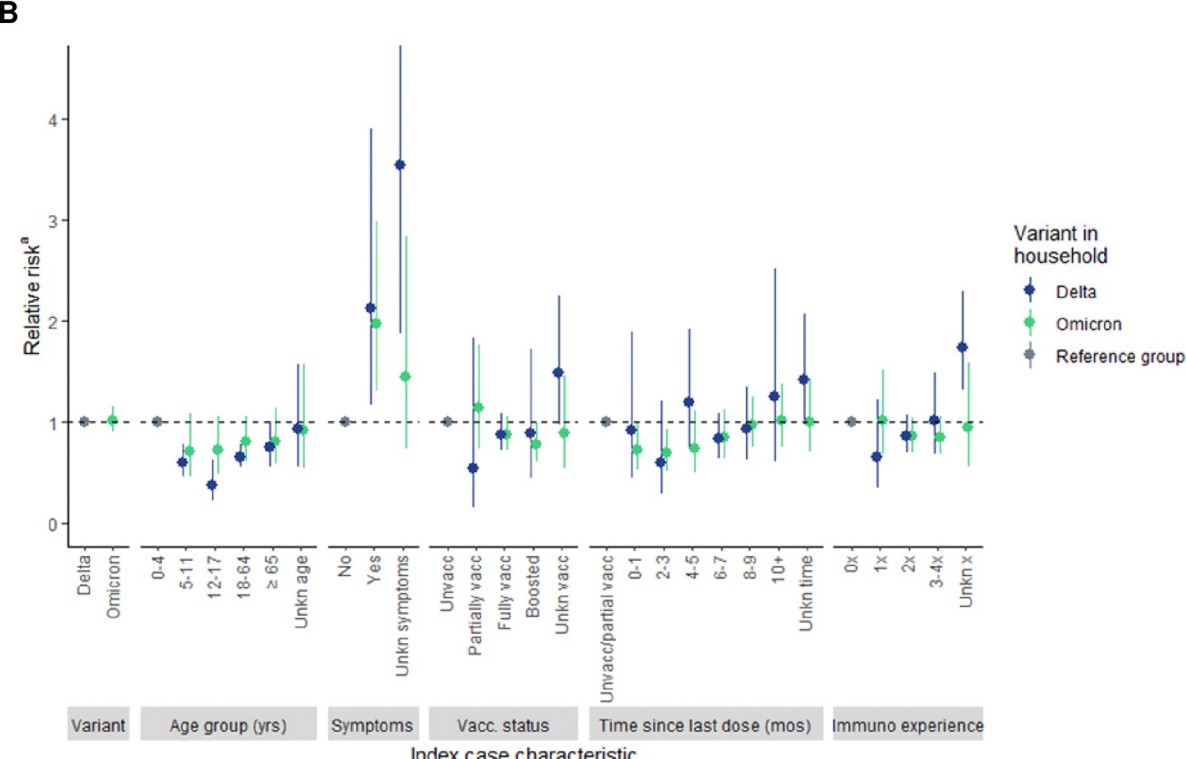

**Fig 3.** A) Attack rate and B) relative risk by index case characteristic and SARS-CoV-2 variant in the household. [a] Y-axis displays a maximum relative risk of 4.5. The upper confidence limit for those with unknown symptom status extends beyond the displayed maximum to 6.7. Abbreviations: Unkn, Unknown; Unvacc, Unvaccinated; Vacc, vaccination; yrs, years; mos, months; Num, Number.

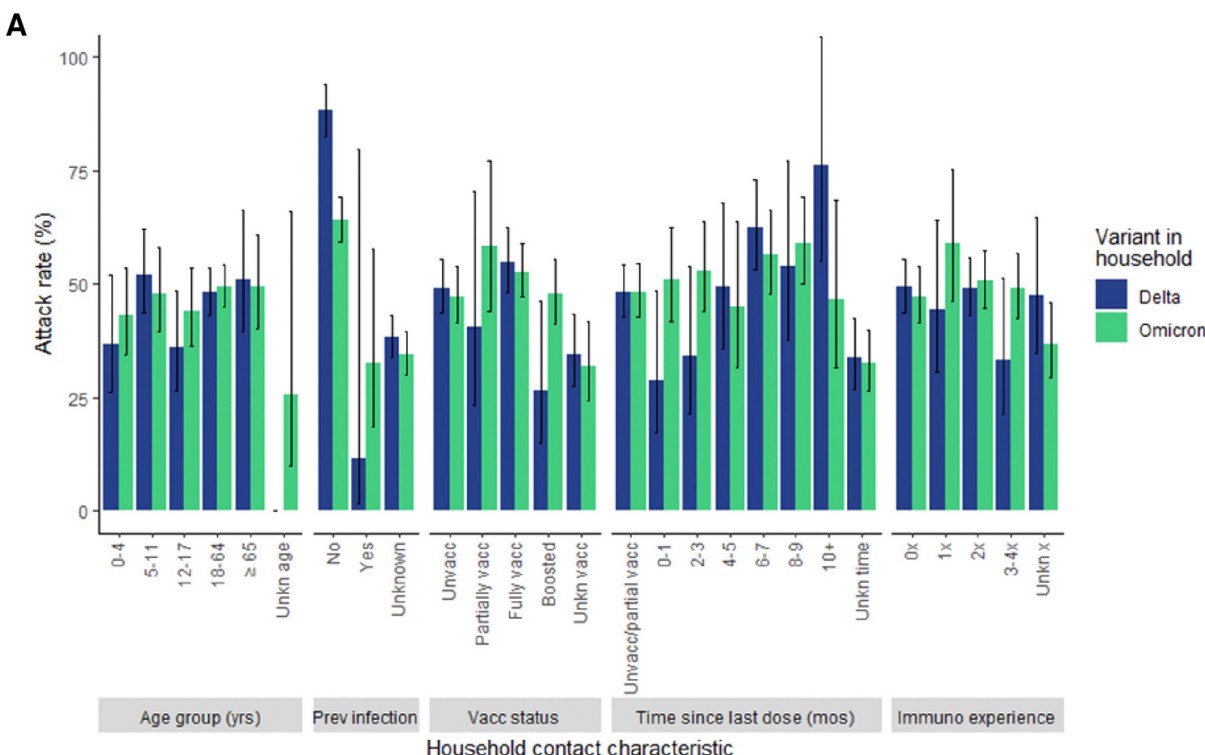

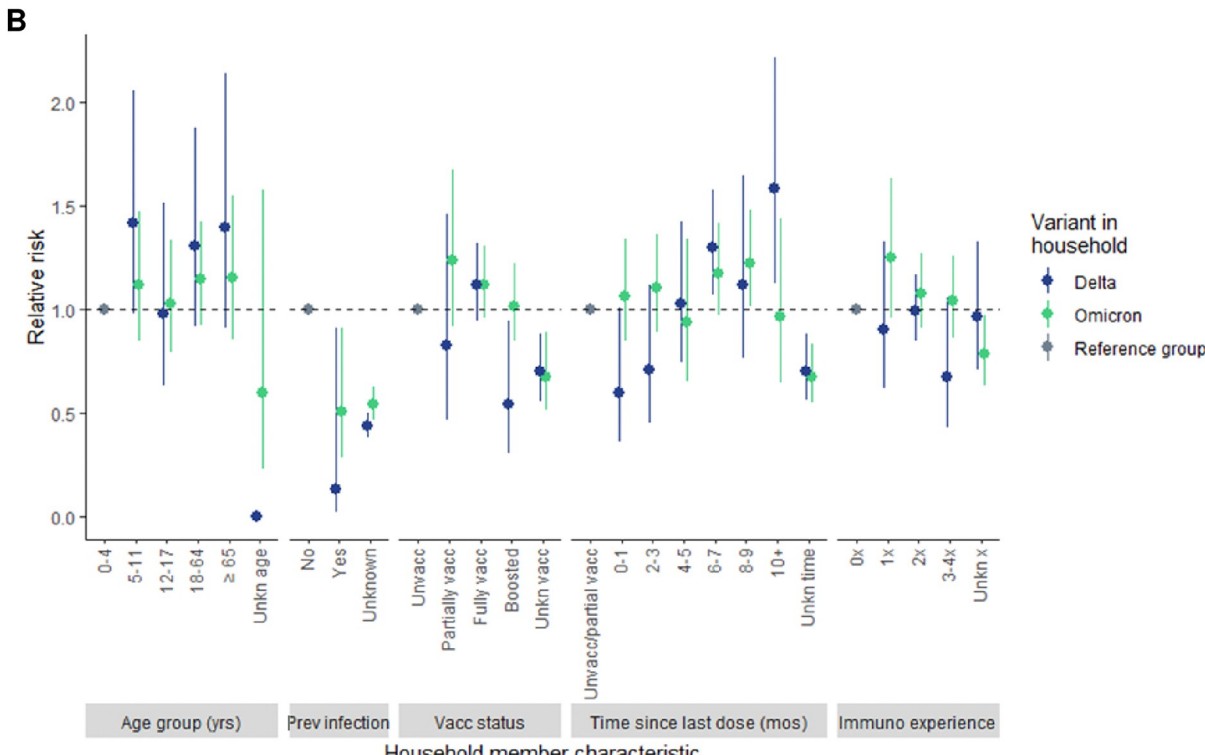

**Fig 4.** A) Attack rate and B) relative risk by household contact characteristic and SARS-CoV-2 variant in the household. Abbreviations: Prev infection, Previous infection; Unkn, Unknown; Unvacc, Unvaccinated; Vacc, Vaccination; yrs, years; mos, months; Immuno experience, number of prior immunologic experiences.

**Table 2. Adjusted relative risk describing SARS-CoV-2 transmission among household contacts controlling for SARS-CoV-2 variant, index case characteristics and household contact characteristics.**

| Characteristic | Expanded/Final model RR (95% CI) N = 1,272 | Sub-analysis[a] RR (95% CI) N = 937 |
|---|---|---|
| **SARS-CoV-2 variant in the household** | | |
| Delta | 1.0 (ref) | 1.0 (ref) |
| Omicron | 1.1 (0.9–1.3) | 1.0 (0.8–1.2) |
| **Index case characteristics** | | |
| **Age group (years)** | | |
| 0–4 | 1.0 (ref) | 1.0 (ref) |
| 5–11 | 0.8 (0.6–1.1) | 0.8 (0.6–1.1) |
| 12–17 | 0.7 (0.5–0.9) | 0.5 (0.3–0.8) |
| 18–64 | 0.8 (0.7–1.0) | 0.7 (0.6–1.0) |
| ≥65 | 0.9 (0.7–1.2) | 0.8 (0.6–1.2) |
| **Symptom status** | | |
| Asymptomatic | 1.0 (ref) | 1.0 (ref) |
| Symptomatic | 2.0 (1.4–2.9) | 2.8 (1.7–4.7) |
| **Vaccination status[b]** | | |
| Unvaccinated/partially vaccinated | 1.0 (ref) | 1.0 (ref) |
| Fully vaccinated | 0.9 (0.8–1.1) | 0.9 (0.7–1.1) |
| Boosted | 0.8 (0.6–1.1) | 0.9 (0.7–1.3) |
| **Household contact characteristics** | | |
| **Age group (years)** | | |
| 0–4 | 1.0 (ref) | 1.0 (ref) |
| 5–11 | 1.3 (1.0–1.7) | 1.5 (1.1–2.0) |
| 12–17 | 1.0 (0.8–1.3) | 1.1 (0.8–1.6) |
| 18–64 | 1.4 (1.1–1.8) | 1.6 (1.1–2.2) |
| ≥65 | 1.5 (1.1–2.2) | 1.9 (1.2–2.9) |
| **Number of prior immunologic experiences[c]** | | |
| 0 | 1.0 (ref) | 1.0 (ref) |
| 1 | 1.0 (0.7–1.3) | 0.8 (0.5–1.2) |
| 2 | 0.9 (0.6–1.3) | 0.8 (0.5–1.3) |
| 3–4 | 0.7 (0.4–1.0) | 0.5 (0.3–0.9) |
| **Time since last vaccine dose (months)- Delta households** | | |
| Unvaccinated/partially vaccinated | 1.0 (ref) | 1.0 (ref) |
| 0–3 | 0.8 (0.5–1.2) | 0.9 (0.5–1.6) |
| 4–7 | 1.3 (0.9–1.8) | 1.4 (0.8–2.4) |
| ≥8 | 1.2 (0.7–1.8) | 1.3 (0.7–2.2) |
| **Time since last vaccine dose (months)- Omicron households** | | |
| Unvaccinated/partially vaccinated | 1.0 (ref) | 1.0 (ref) |
| 0–3 | 1.2 (0.8–1.7) | 1.1 (0.5–2.2) |
| 4–7 | 1.1 (0.8–1.5) | 1.0 (0.5–2.0) |

(*Continued*)

**Table 2.** (Continued)

| Characteristic | Expanded/Final model RR (95% CI) N = 1,272 | Sub-analysis[a] RR (95% CI) N = 937 |
|---|---|---|
| ≥8 | 1.1 (0.8–1.5) | 0.9 (0.4–1.7) |

Abbreviations: Ref, reference group; RR, relative risk; CI, confidence interval

[a] In the sub-analysis, the final model was applied to the subset of households which were identified retrospectively.

[b] Vaccination status was defined as of the index case date and was categorized as unvaccinated (never received a COVID-19 vaccine), partially vaccinated (completed part of a COVID-19 vaccine series or completed the primary series <14 days prior), fully vaccinated (completed the primary series ≥14 days prior or received a booster dose <7 days prior), or boosted (received an additional vaccine dose beyond the primary series ≥7 days prior).

[c] Number of prior immunologic experiences combined previous infection and vaccination doses to estimate the number of times a person's immune system had been exposed to SARS-CoV-2 or the vaccine. The possible range of values was zero to four exposures and could be any combination of infection and vaccination. Those with unknown previous infection status were considered to have no previous infection.

symptom status, with transmission to household contacts being twice as likely (RR = 2.0, 95% CI: 1.4–2.9) when the index case was symptomatic compared to asymptomatic. In Delta households, those 0–3 months from their last vaccine dose were at the lowest risk of infection (RR = 0.8, 95% CI: 0.5–1.2) and those 4–7 or ≥8 months were at increased risk, relative to unvaccinated or partially vaccinated household contacts, though confidence intervals overlapped; in Omicron households, risk was relatively consistent for all groups. Some variation in the RRs for both age of the index case and age of the household contact were observed, with risk being lowest when the index case was 12–17 years of age (RR = 0.7, 95% CI: 0.5–0.9) and greatest when the household contact was 18 years of age or older (RR for 18–64 year age group = 1.4, 95% CI: 1.1–1.8; RR for ≥65 year age group = 1.5, 95% CI: 1.1–2.2), with overlapping confidence intervals. Expected patterns were observed for other index case and household contact characteristics but with wide confidence intervals that included the null.

When the final model was applied to the subset of retrospectively identified households, the RRs changed only modestly (Table 2). The most substantial difference between the models was observed for index case symptom status. In the retrospective subset, the RR increased to 2.8 (95% CI: 1.7–4.7).

## Discussion

In the complicated epidemiologic setting where biological and social factors combine to impact SARS-CoV-2 transmission, this study demonstrates that few factors strongly influenced transmission risk in households. After accounting for several index case and household contact characteristics using an adjusted multivariable model, only time since household contact's last vaccine dose seemed to differ by variant with an attenuated impact observed in Omicron households when compared to Delta households. For both variants, index case symptom status was the only strong predictor of transmission risk. These findings emphasize the importance of adjusted models that account for the multifactorial drivers of transmission and the importance of symptom status when predicting transmissibility of currently circulating Omicron sub-lineages or emerging strains.

SARS-CoV-2 variants and sub-lineages have exhibited distinct variation in traits, such as the ability to evade immune defenses or cause severe disease, requiring ongoing assessment of the public health and clinical impact of SARS-CoV-2 over time. Much as household-based studies have elucidated the epidemiology of the ever-changing influenza virus over the last

several decades [11, 12], they can provide early insight into the transmission of SARS-CoV-2 as the virus evolves. Important strengths of household-based studies are the well-defined cohorts exposed to the index case and the relatively short duration of follow-up required to identify infection [11], facilitating rapid evaluation of emerging variants. The current study adds to the growing body of literature comparing transmission of Omicron to previously circulating variants [5, 7, 13] and provides baseline data from the early months of Omicron's emergence for comparison to currently circulating Omicron sub-lineages [14], contributing to our understanding of collective and cumulative changes that occur over the course of the pandemic.

This unique analysis emphasizes the importance of accounting for multiple index case and household contact characteristics when determining transmission risk by variant. In unadjusted analyses, several characteristics, including previous infection status, displayed expected risk patterns in Delta households and resembled features of endemic coronavirus, such as short-term protection against reinfection [15, 16] and building immunity with repeat exposures [17]; these patterns were attenuated among household contacts in Omicron households suggesting potential differences by variant. In contrast, household contact time since the last vaccine dose was the only characteristic with a significant interaction term in exploration of adjusted analyses. Overall, the adjusted results estimate only slight changes in risk factors for infection among household contacts with the shift from Delta to Omicron, suggesting that the observed differences between the Delta and Omicron waves of the pandemic may have been more related to extra-household transmission factors (e.g. reduced mitigation measures, seasonal differences in human behavior, or viral factors). Extrapolating to the more recently circulating Omicron sub-lineages [14], we might expect only small changes in risk given the close genetic relationship of the circulating sub-lineages to the Omicron lineage investigated here and the ongoing COVID-19 vaccination and infection occurring in the population. Whether future variants result in more major changes to risk will likely depend on the timing and magnitude of SARS-CoV-2 evolution and the immunologic implications of the variant mutations [18, 19].

The minimal change in transmission risk observed with the shift from Delta to Omicron may be related to the nature of household exposure, which is likely to be prolonged, close contact exposure to an index case. Factors such as immune evasiveness [5] and susceptibility of the population, measured in this investigation via household contact time since last vaccination dose and the cumulative number of prior immunologic experiences, may play a smaller role in household transmission where exposure is high. While both the Delta and Omicron ARs estimated in this investigation lay within the range of ARs described in a meta-analysis of household transmission studies (Delta: 9%-63%; Omicron: 31%-53%) [7], the median household size of 3 (IQR: 2–4) may reduce the generalizability of these findings to larger households.

In both Delta and Omicron households, we found household contacts of symptomatic index cases to be twice as likely to become infected compared to contacts of asymptomatic index cases. This strong, consistent association may be representative of viral and individual characteristics. Individuals infected with newer SARS-CoV-2 variants tend to have higher viral loads [20] and emit more airborne virus relative to those infected with variants that circulated early in the COVID-19 pandemic [21]. There is not strong evidence, however, of differences in COVID-19 disease symptomatology in recent variants [22], which may account for the lack of association between symptom status and variant in this analysis. Within a host, increased individual-level immunity from vaccination and/or previous infections may result in reduced severity or duration of symptoms [23, 24] that, in turn, reduce the infectiousness of the individual. Interestingly, index case vaccination status was not found to be significantly associated

with transmission when included in the adjusted model; however, this may be because the impact of vaccination was partly accounted for in symptom status.

Several limitations should be noted. Data analysis assumed equal SARS-CoV-2 exposure among household contacts of the index cases and did not account for socioeconomic status, behavioral characteristics such as infection control precautions, or household crowding, which may substantially influence risk [25, 26]. While households with individuals who reported known exposures to SARS-CoV-2 (other than the index case) were excluded from the analysis, this information was unavailable for retrospectively identified households. SARS-CoV-2 testing was not conducted systematically and at-home testing information may be incomplete. To minimize potential bias and improve case ascertainment, case status determination incorporated symptom status in addition to COVID-19 test results. It is still possible, however, that some cases were missed if testing and/or symptom onset occurred after the household was interviewed, potentially resulting in underestimation of ARs. Date of vaccination doses and previous infections were not available for all index case and household contacts, reducing the sample size for time-specific factors (e.g. time since last vaccine dose) and potentially contributing to misclassification. Data collection methods for prospectively and retrospectively identified households differed and may have biased estimates if one method was more or less likely to accurately identify cases among household members. A sub-analysis restricted to retrospectively identified households resulted in RRs similar to those using the full dataset, suggesting such bias was likely limited. Lastly, SARS-CoV-2 sequencing data was typically only available for one member of the household, preventing confirmation that household contacts were infected by the index case.

## Conclusion

This investigation quantified the contribution of several index case and household contact characteristics to transmission risk in households with Delta or Omicron circulating and, uniquely, applied a multivariable model to produce estimates reflecting the complex relationships between variant, index case characteristics, and household contact characteristics. After adjusting for multiple factors, only time since household contact's last COVID-19 vaccine dose was observed to differ modestly by variant and few factors substantially impacted risk. Symptom status of the index case was the only consistent and strong predictor of household transmission regardless of variant and may serve as an important clinical characteristic to consider when assessing the public health impact of emerging strains. As SARS-CoV-2 evolves [14], household transmission studies using multivariable modeling approaches can serve as rapid and ongoing sources of information on changes in transmission risk in the increasingly complex epidemiologic setting.

## Supporting information

**S1 Table. Models and interaction terms assessed to estimate the adjusted relative risk describing SARS-CoV-2 transmission among household contacts controlling for SARS-CoV-2 variant, index case characteristics and household contact characteristics.** Abbreviations: Ref, reference group; RR, relative risk; CI, confidence interval, [a] Final model shown in Table 2. [b] Data shown are the exponentiated interaction terms (ratio of Delta:Omicron relative risks). An interaction term was included in the final model if the term was significant at the α = 0.05 level (i.e. the 95% confidence interval did not include the null). For multilevel variables, an interaction term was included if it was significant at the α = 0.05 level for any level of the variable. [c] Vaccination status was defined as of the index case date and was categorized as unvaccinated (never received a COVID-19 vaccine), partially vaccinated (completed part of a

COVID-19 vaccine series or completed the primary series <14 days prior), fully vaccinated (completed the primary series ≥14 days prior or received a booster dose <7 days prior), or boosted (received an additional vaccine dose beyond the primary series ≥7 days prior). [d] Number of prior immunologic experiences combined previous infection and vaccination doses to estimate the number of times a person's immune system had been exposed to SARS-CoV-2 or the vaccine. The possible range of values was zero to four exposures and could be any combination of infection and vaccination. Those with unknown previous infection status were considered to have no previous infection for this composite variable.
(DOCX)

## Acknowledgments

We thank the investigation's participants who contributed their time and information. We also thank: Alexandra Mellis, Melissa Rolfes, Phillip Salvatore, Olivia Almendares, Sarah E. Smith-Jeffcoat, Emeka Oraka, CDC; Charles Powell, Avery Gartman, Connecticut Department of Public Health; Carla Barrios, Alexandria Davis, Christine Roloff, Ashley Becht, Hallie Hutchinson, Eugene Olshansky, Rachel Berg, Adrianna Koczwara, Lisa Addis, Michael Deneufbourg, Sara Love, Isaac Ghinai, Peter Ruestow, Shamika Smith, Daniel Liguori, Frances Lendacki, Janna Kerins, Stephanie Black, Chicago Department of Public Health; Stefan Green, Hannah Barbian, Sofiya Bobrovska, Alyse Kittner, Regional Innovative Public Health Laboratory, Chicago Department of Public Health and Rush University Medical Center; Lindsey Page, Barbara Cuene, Stephen Fendt, Jennifer Lares, Carri Marlow, Nandhu Balakrishnan, Katherine Akinyemi, Addie Skillman, Milwaukee Health Department; Leisha Nolen, Eric Walls, Sia Gerard, Megan Tippetts, Alexis Molina, Shai Miguel, Alix Elliston, April Jorgensen, Indigo Newbold, Garnet Kwader, Sam Andersen, Utah Department of Health and Human Services.

**Disclaimer:** The findings and conclusions in this report are those of the author(s) and do not necessarily represent the official position of the Centers for Disease Control and Prevention.

## Author Contributions

**Conceptualization:** Hannah L. Kirking, Jacqueline E. Tate.

**Data curation:** Julia M. Baker, Jasmine Y. Nakayama, Michelle O'Hegarty, Andrea McGowan, Richard A. Teran, Stephen M. Bart, Lynn E. Sosa, Jessica Brockmeyer, Kayla English, Katie Mosack, Alina Paegle, Thelonious W. Williams.

**Formal analysis:** Julia M. Baker, Jasmine Y. Nakayama.

**Investigation:** Julia M. Baker, Jasmine Y. Nakayama, Michelle O'Hegarty, Andrea McGowan, Richard A. Teran, Stephen M. Bart, Lynn E. Sosa, Jessica Brockmeyer, Kayla English, Katie Mosack, Sanjib Bhattacharyya, Manjeet Khubbar, Brooke Campos, Alina Paegle, John McGee, Thelonious W. Williams.

**Methodology:** Julia M. Baker, Jasmine Y. Nakayama, Manjeet Khubbar.

**Project administration:** Sanjib Bhattacharyya, Nicole R. Yerkes, Brooke Campos, Alina Paegle, Robert Herrera, Hannah L. Kirking, Jacqueline E. Tate.

**Supervision:** Katie Mosack, Nicole R. Yerkes, Brooke Campos, Alina Paegle, John McGee, Robert Herrera, Marcia Pearlowitz, Hannah L. Kirking, Jacqueline E. Tate.

**Visualization:** Julia M. Baker.

**Writing – original draft:** Julia M. Baker.

**Writing – review & editing:** Julia M. Baker, Jasmine Y. Nakayama, Michelle O'Hegarty, Andrea McGowan, Richard A. Teran, Stephen M. Bart, Lynn E. Sosa, Jessica Brockmeyer, Kayla English, Katie Mosack, Sanjib Bhattacharyya, Manjeet Khubbar, Alina Paegle, Marcia Pearlowitz, Thelonious W. Williams, Hannah L. Kirking, Jacqueline E. Tate.

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
