## [Decision Letter · Decision Letter 0]

20 Aug 2024

PONE-D-24-00274Household transmission of SARS-CoV-2 in five US jurisdictions: comparison of Delta and Omicron variantsPLOS ONE

Dear Dr. Baker,

Thank you for submitting your manuscript to PLOS ONE. After careful consideration, we feel that it has merit but does not fully meet PLOS ONE’s publication criteria as it currently stands. Therefore, we invite you to submit a revised version of the manuscript that addresses the points raised during the review process.

We look forward to receiving your revised manuscript.

Kind regards,

Ranjan K. Mohapatra, PhD

Academic Editor

PLOS ONE

“I have read the journal's policy and the authors of this manuscript have the following competing interests:

LES: Support for the present manuscript from Centers for Disease Control and Prevention; Support for attending meetings and/or travel from Centers for Disease Control and Prevention, Council of State and Territorial Epidemiologists, Association of State and Territorial Health Officers; Participation on a Data Safety Monitoring Board or Advisory Board for Centers for Disease Control and Prevention.

NRY: Grants or contracts from CDC’s Epidemiology and Laboratory Capacity for Prevention and Control of Emerging Infectious Diseases (ELC) Cares, ELC PPP, ELC EED (Constituted hourly wages received through Utah Department of Health and Human Services’ payroll system).

BC: Grants or contracts from CDC’s Epidemiology and Laboratory Capacity for Prevention and Control of Emerging Infectious Diseases (ELC) PPPHEA, ELC ED (Constituted hourly wages received through Utah Department of Health and Human Services’ payroll system).

AP: Grants or contracts from CDC’s Epidemiology and Laboratory Capacity for Prevention and Control of Emerging Infectious Diseases (ELC) Cares, ELC PPP, ELC EED (Constituted hourly wages received through Utah Department of Health and Human Services’ payroll system).

RH: Support for the present manuscript from CDC’s Epidemiology and Laboratory Capacity for Prevention and Control of Emerging Infectious Diseases (ELC) Cares (Constituted hourly wages and received through Utah Department of Health and Human Services’ payroll system); Grants or contracts from ELC PPP and Immunization Cooperative Agreements (Constituted hourly wages and received through Utah Department of Health and Human Services’ payroll system).

MP: Support for the present manuscript from CDC’s Epidemiology and Laboratory Capacity for Prevention and Control of Emerging Infectious Diseases (ELC) (Cooperative agreement funding supported staff to conduct routine disease investigations and data analysis).

TWW: Support for the present manuscript from DC Grant: CDC’s Epidemiology and Laboratory Capacity for Prevention and Control of Emerging Infectious Diseases (ELC) (Cooperative agreement funding supported staff to conduct routine disease investigations and data analysis).”

3. For studies involving third-party data, we encourage authors to share any data specific to their analyses that they can legally distribute. PLOS recognizes, however, that authors may be using third-party data they do not have the rights to share. When third-party data cannot be publicly shared, authors must provide all information necessary for interested researchers to apply to gain access to the data. (https://journals.plos.org/plosone/s/data-availability#loc-acceptable-data-access-restrictions)

a) A description of the data set and the third-party source

b) If applicable, verification of permission to use the data set

c) Confirmation of whether the authors received any special privileges in accessing the data that other researchers would not have

d) All necessary contact information others would need to apply to gain access to the data

Additional Editor Comments:

I have received the comments from the reviewers. However, it may further be considered with the following changes.

1. Authors should discuss the novelty of the work in last part of abstract and conclusion.

2. The study was carried out from October 2021 through February 2022 in the US. However, authors have not discussed the application of this work in 2024.

3. Discussion part must be improved.

4. Add limitations of the study after discussion.

5. The important findings of the study should be discussed in conclusion section.

Reviewers' comments:

Reviewer's Responses to Questions

**Comments to the Author**

1. Is the manuscript technically sound, and do the data support the conclusions?

Reviewer #1: Yes

Reviewer #2: Yes

2. Has the statistical analysis been performed appropriately and rigorously? 

Reviewer #1: Yes

Reviewer #2: Yes

3. Have the authors made all data underlying the findings in their manuscript fully available?

Reviewer #1: Yes

Reviewer #2: Yes

4. Is the manuscript presented in an intelligible fashion and written in standard English?

Reviewer #1: Yes

Reviewer #2: Yes

5. Review Comments to the Author

Reviewer #1: The manuscript is technically sound and the analyzed and interpreted data supported the conclusion. Experiments were conducted with appropriate sample size of replication and control. In addition statistics data was done properly and sketched in graphics. The manuscript write-up is written in clearly with no grammatical error.

Reviewer #2: The manuscript offers a significant contribution to the field of public health, addressing an important global health issue. The authors present a well-structured study that is methodologically sound, and theoretically grounded and contributes new insights. I recommend that the paper be accepted for publication.

6. PLOS authors have the option to publish the peer review history of their article (what does this mean?). If published, this will include your full peer review and any attached files.

Reviewer #1: **Yes: **Asefa Deressa, Research Adviser

Reviewer #2: **Yes: **Lawrence Sena Tuglo

---

## [Author Response · Author response to Decision Letter 0]

21 Oct 2024

Please note: Responses to the editor/reviewers are in blue and line numbers refer to the track changes version of the manuscript.

Editor Comments:

I have received the comments from the reviewers. However, it may further be considered with the following changes.

1. Authors should discuss the novelty of the work in last part of abstract and conclusion.

Response: 

Thank you for your suggestion to better highlight the novelty of this work. In both the abstract and conclusion sections, we have elaborated on how this investigation is unique in its use of a multivariable model to account for the interaction between various index case and household contact characteristics and their role in household transmission risk. This approach more accurately captures the combined role of multiple predictors which contribute to risk in today’s COVID-19 environment. 

Updates to text: 

(lines 53-56) “Uniquely, this study adjusted risk estimates for several index case and household contact characteristics and demonstrates that few characteristics strongly dictate risk, likely reflecting the complexity of the biological and social factors which combine to impact SARS-CoV-2 transmission.”

(lines 443-447) “This investigation quantified the contribution of several index case and household contact characteristics to transmission risk in households with Delta or Omicron circulating and, uniquely, applied a multivariable model to produce estimates reflecting the complex relationships between variant, index case characteristics, and household contact characteristics.

2. The study was carried out from October 2021 through February 2022 in the US. However, authors have not discussed the application of this work in 2024.

Response:

Throughout the discussion and conclusion sections, we have added text to elaborate on how this investigation applies to today’s COVID-19 environment and currently circulating strains. We explain that 1) the household transmission study approach will remain useful for rapid evaluation of transmission as SARS-CoV-2 evolves; 2) the patterns observed in this evaluation in the Delta and early Omicron period likely hold true today as Omicron is still circulating (albeit slightly differently as new sub-variants have emerged) and COVID-19 vaccination and infection still occur, contributing to multifactorial components of transmission risk; and 3) the primary predictor of transmission identified in this study, index case symptom status, should be a key characteristic evaluated in current and future variants when estimating their public health impact. Relatedly, we have added several new references describing the importance of household transmission studies for understanding infectious disease transmission (ref #11-12), information on recently circulating SARS-CoV-2 strains (ref #14), and viral characteristics that may be related to symptom status of infections (ref #20-24)

Updates to text: 

(lines 346-349) “These findings emphasize the importance of adjusted models that account for the multifactorial drivers of transmission and the importance of symptom status when predicting transmissibility of currently circulating Omicron sub-lineages or emerging strains.”

(lines 351-362) “SARS-CoV-2 variants and sub-lineages have exhibited distinct variation in traits, such as the ability to evade immune defenses or cause severe disease, requiring ongoing assessment of the public health and clinical impact of SARS-CoV-2 over time. Much as household-based studies have elucidated the epidemiology of the ever-changing influenza virus over the last several decades [11,12], they can provide early insight into the transmission of SARS-CoV-2 as the virus evolves. Important strengths of household-based studies are the well-defined cohorts exposed to the index case and the relatively short duration of follow-up required to identify infection [11], facilitating rapid evaluation of emerging variants. The current study adds to the growing body of literature comparing transmission of Omicron to previously circulating variants [5,7,13] and provides baseline data from the early months of Omicron’s emergence for comparison to currently circulating Omicron sub-lineages [14], contributing to our understanding of collective and cumulative changes that occur over the course of the pandemic.”

(lines 386-389) “Extrapolating to the more recently circulating Omicron sub-lineages [14], we might expect only small changes in risk given the close genetic relationship of the circulating sub-lineages to the Omicron lineage investigated here and the ongoing COVID-19 vaccination and infection occurring in the population.”

(lines 452-454) “Symptom status of the index case was the only consistent and strong predictor of household transmission regardless of variant and may serve as an important clinical characteristic to consider when assessing the public health impact of emerging strains.”

3. Discussion part must be improved.

Response:

Thank you for the suggestion to enhance the discussion section. We have substantially modified the text throughout to address the other comments received (please see other responses). In addition, we have included two new paragraphs which detail the benefit of household transmission studies and interpret the finding that transmission is substantially higher when the index case is symptomatic. We feel the discussion section now highlights and interprets the results of this investigation more completely. 

Updates to text: 

(lines 351-362) “SARS-CoV-2 variants and sub-lineages have exhibited distinct variation in traits, such as the ability to evade immune defenses or cause severe disease, requiring ongoing assessment of the public health and clinical impact of SARS-CoV-2 over time. Much as household-based studies have elucidated the epidemiology of the ever-changing influenza virus over the last several decades [11,12], they can provide early insight into the transmission of SARS-CoV-2 as the virus evolves. Important strengths of household-based studies are the well-defined cohorts exposed to the index case and the relatively short duration of follow-up required to identify infection [11], facilitating rapid evaluation of emerging variants. The current study adds to the growing body of literature comparing transmission of Omicron to previously circulating variants [5,7,13] and provides baseline data from the early months of Omicron’s emergence for comparison to currently circulating Omicron sub-lineages [14], contributing to our understanding of collective and cumulative changes that occur over the course of the pandemic.”

(lines 405-417) “In both Delta and Omicron households, we found household contacts of symptomatic index cases to be twice as likely to become infected compared to contacts of asymptomatic index cases. This strong, consistent association may be representative of viral and individual characteristics. Individuals infected with newer SARS-CoV-2 variants tend to have higher viral loads [20] and emit more airborne virus relative to those infected with variants that circulated early in the COVID-19 pandemic [21]. There is not strong evidence, however, of differences in COVID-19 disease symptomatology in recent variants [22], which may account for the lack of association between symptom status and variant in this analysis. Within a host, increased individual-level immunity from vaccination and/or previous infections may result in reduced severity or duration of symptoms [23,24] that, in turn, reduce the infectiousness of the individual. Interestingly, index case vaccination status was not found to be significantly associated with transmission when included in the adjusted model; however, this may be because the impact of vaccination was partly accounted for in symptom status.”

4. Add limitations of the study after discussion.

Response:

A substantial limitations section is already included in the discussion section, however, we have demarcated it with a new sub-header, “Limitations” to distinguish it from the rest of the text.

Updates to text: 

(line 419) “Limitations”

5. The important findings of the study should be discussed in conclusion section.

Response:

We have more clearly highlighted the study’s main findings in the conclusion section and clarified the impact of these findings on COVID-19 today and into the future as the virus evolves. 

Updates to text: 

(lines 450-458) “After adjusting for multiple factors, only time since household contact’s last COVID-19 vaccine dose was observed to differ modestly by variant and few factors substantially impacted risk. Symptom status of the index case was the only consistent and strong predictor of household transmission regardless of variant and may serve as an important clinical characteristic to consider when assessing the public health impact of emerging strains.”

Reviewers' comments:

1. Is the manuscript technically sound, and do the data support the conclusions?

Reviewer #1: Yes

Reviewer #2: Yes

Response: Thank you.

2. Has the statistical analysis been performed appropriately and rigorously? 

Reviewer #1: Yes

Reviewer #2: Yes

Response: Thank you.

3. Have the authors made all data underlying the findings in their manuscript fully available?

Reviewer #1: Yes

Reviewer #2: Yes

Response: Thank you.

4. Is the manuscript presented in an intelligible fashion and written in standard English?

Reviewer #1: Yes

Reviewer #2: Yes

Response: Thank you.

5. Review Comments to the Author

Reviewer #1: The manuscript is technically sound and the analyzed and interpreted data supported the conclusion. Experiments were conducted with appropriate sample size of replication and control. In addition statistics data was done properly and sketched in graphics. The manuscript write-up is written in clearly with no grammatical error.

Response: We appreciate your thoughtful evaluation.

Reviewer #2: The manuscript offers a significant contribution to the field of public health, addressing an important global health issue. The authors present a well-structured study that is methodologically sound, and theoretically grounded and contributes new insights. I recommend that the paper be accepted for publication.

Response: Thank you for your review.

6. PLOS authors have the option to publish the peer review history of their article (what does this mean?). If published, this will include your full peer review and any attached files.

Do you want your identity to be public for this peer review? For information about this choice, including consent withdrawal, please see our Privacy Policy.

Reviewer #1: Yes: Asefa Deressa, Research Adviser

Reviewer #2: Yes: Lawrence Sena Tuglo

Response: Thank you very much for your time and consideration.

---

## [Editor Report · Decision Letter 1]

30 Oct 2024

Household transmission of SARS-CoV-2 in five US jurisdictions: comparison of Delta and Omicron variants

PONE-D-24-00274R1

Dear Dr. Baker,

We’re pleased to inform you that your manuscript has been judged scientifically suitable for publication and will be formally accepted for publication once it meets all outstanding technical requirements.

Kind regards,

Ranjan K. Mohapatra, PhD

Academic Editor

PLOS ONE

Additional Editor Comments (optional):

Authors have improved the manuscript as suggested. It may be accepted for publication after plagiarism check.
---

## [Editor Report · Acceptance letter]

5 Nov 2024

PONE-D-24-00274R1 

PLOS ONE

Dear Dr. Baker, 

I'm pleased to inform you that your manuscript has been deemed suitable for publication in PLOS ONE. Congratulations! Your manuscript is now being handed over to our production team.

Kind regards, 

on behalf of

Dr. Ranjan K. Mohapatra 

Academic Editor

PLOS ONE